# Detecting Statistical Interactions From Neural Network Weights

**Michael Tsang, Dehua Cheng, Yan Liu**
Department of Computer Science
University of Southern California
`{tsangm,dehuache,yanliu.cs}@usc.edu`

## Abstract

Interpreting neural networks is a crucial and challenging task in machine learning. In this paper, we develop a novel framework for detecting statistical interactions captured by a feedforward multilayer neural network by directly interpreting its learned weights. Depending on the desired interactions, our method can achieve significantly better or similar interaction detection performance compared to the state-of-the-art without searching an exponential solution space of possible interactions. We obtain this accuracy and efficiency by observing that interactions between input features are created by the non-additive effect of nonlinear activation functions, and that interacting paths are encoded in weight matrices. We demonstrate the performance of our method and the importance of discovered interactions via experimental results on both synthetic datasets and real-world application datasets.

## 1 Introduction

Despite their strong predictive power, neural networks have traditionally been treated as "black box" models, preventing their adoption in many application domains. It has been noted that complex machine learning models can learn unintended patterns from data, raising significant risks to stakeholders (Varshney & Alemzadeh, 2016). Therefore, in applications where machine learning models are intended for making critical decisions, such as healthcare or finance, it is paramount to understand how they make predictions (Caruana et al., 2015; Goodman & Flaxman, 2016).

Existing approaches to interpreting neural networks can be summarized into two types. One type is direct interpretation, which focuses on 1) explaining individual feature importance, for example by computing input gradients (Simonyan et al., 2013; Ross et al., 2017; Sundararajan et al., 2017) or by decomposing predictions (Bach et al., 2015; Shrikumar et al., 2017), 2) developing attention-based models, which illustrate where neural networks focus during inference (Itti et al., 1998; Mnih et al., 2014; Xu et al., 2015), and 3) providing model-specific visualizations, such as feature map and gate activation visualizations (Yosinski et al., 2015; Karpathy et al., 2015). The other type is indirect interpretation, for example post-hoc interpretations of feature importance (Ribeiro et al., 2016) and knowledge distillation to simpler interpretable models (Che et al., 2016).

It has been commonly believed that one major advantage of neural networks is their capability of modeling complex statistical interactions between features for automatic feature learning. Statistical interactions capture important information on where features often have joint effects with other features on predicting an outcome. The discovery of interactions is especially useful for scientific discoveries and hypothesis validation. For example, physicists may be interested in understanding what joint factors provide evidence for new elementary particles; doctors may want to know what interactions are accounted for in risk prediction models, to compare against known interactions from existing medical literature.

In this paper, we propose an accurate and efficient framework, called *Neural Interaction Detection* (`NID`), which detects statistical interactions of any order or form captured by a feedforward neural network, by examining its weight matrices. Our approach is efficient because it avoids searching over an exponential solution space of interaction candidates by making an approximation of hidden unit importance at the first hidden layer via all weights above and doing a 2D traversal of the input

weight matrix. We provide theoretical justifications on why interactions between features are created at hidden units and why our hidden unit importance approximation satisfies bounds on hidden unit gradients. Top-$K$ true interactions are determined from interaction rankings by using a special form of generalized additive model, which accounts for interactions of variable order (Wood, 2006; Lou et al., 2013). Experimental results on simulated datasets and real-world datasets demonstrate the effectiveness of `NID` compared to the state-of-the-art methods in detecting statistical interactions.

The rest of the paper is organized as follows: we first review related work and define notations in Section 2. In Section 3, we examine and quantify the interactions encoded in a neural network, which leads to our framework for interaction detection detailed in Section 4. Finally, we study our framework empirically and demonstrate its practical utility on real-world datasets in Section 5.

## 2 RELATED WORK AND NOTATIONS

### 2.1 INTERACTION DETECTION

Statistical interaction detection has been a well-studied topic in statistics, dating back to the 1920s when two-way ANOVA was first introduced (Fisher, 1925). Since then, two general approaches emerged for conducting interaction detection. One approach has been to conduct individual tests for each combination of features (Lou et al., 2013). The other approach has been to pre-specify all interaction forms of interest, then use lasso to simultaneously select which are important (Tibshirani, 1996; Bien et al., 2013; Min et al., 2014; Purushotham et al., 2014).

Notable approaches such as *ANOVA* and *Additive Groves* (Sorokina et al., 2008) belong to the first group. Two-way ANOVA has been a standard method of performing pairwise interaction detection that involves conducting hypothesis tests for each interaction candidate by checking each hypothesis with F-statistics (Wonnacott & Wonnacott, 1972). Besides two-way ANOVA, there is also three-way ANOVA that performs the same analyses but with interactions between three variables instead of two; however, four-way ANOVA and beyond are rarely done because of how computationally expensive such tests become. Specifically, the number of interactions to test grows exponentially with interaction order.

*Additive Groves* is another method that conducts individual tests for interactions and hence must face the same computational difficulties; however, it is special because the interactions it detects are not constrained to any functional form e.g. multiplicative interactions. The unconstrained manner by which interactions are detected is advantageous when the interactions are present in highly nonlinear data (Sorokina et al., 2007; 2008). *Additive Groves* accomplishes this by comparing two regression trees, one that fits all interactions, and the other that has the interaction of interest forcibly removed.

In interaction detection, lasso-based methods are popular in large part due to how quick they are at selecting interactions. One can construct an additive model with many different interaction terms and let lasso shrink the coefficients of unimportant terms to zero (Tibshirani, 1996). While lasso methods are fast, they require specifying all interaction terms of interest. For pairwise interaction detection, this requires $O(p^2)$ terms (where $p$ is the number of features), and $O(2^p)$ terms for higher-order interaction detection. Still, the form of interactions that lasso-based methods capture is limited by which are pre-specified.

Our approach to interaction detection is unlike others in that it is both fast and capable of detecting interactions of variable order without limiting their functional forms. The approach is fast because it does not conduct individual tests for each interaction to accomplish higher-order interaction detection. This property has the added benefit of avoiding a high false positive-, or *false discovery rate*, that commonly arises from multiple testing (Benjamini & Hochberg, 1995).

### 2.2 INTERPRETABILITY

The interpretability of neural networks has largely been a mystery since their inception; however, many approaches have been developed in recent years to interpret neural networks in their traditional feedforward form and as deep architectures. Feedforward neural networks have undergone multiple advances in recent years, with theoretical works justifying the benefits of neural network depth (Telgarsky, 2016; Liang & Srikant, 2016; Rolnick & Tegmark, 2018) and new research on interpreting

feature importance from input gradients (Hechtlinger, 2016; Ross et al., 2017; Sundararajan et al., 2017). Deep architectures have seen some of the greatest breakthroughs, with the widespread use of attention mechanisms in both convolutional and recurrent architectures to show where they focus on for their inferences (Itti et al., 1998; Mnih et al., 2014; Xu et al., 2015). Methods such as feature map visualization (Yosinski et al., 2015), de-convolution (Zeiler & Fergus, 2014), saliency maps (Simonyan et al., 2013), and many others have been especially important to the vision community for understanding how convolutional networks represent images. With long short-term memory networks (LSTMs), a research direction has studied multiplicative interactions in the unique gating equations of LSTMs to extract relationships between variables across a sequence (Arras et al., 2017; Murdoch et al., 2018).

Unlike previous works in interpretability, our approach extracts generalized non-additive interactions between variables from the weights of a neural network.

## 2.3 NOTATIONS

Vectors are represented by boldface lowercase letters, such as $\mathbf{x}, \mathbf{w}$; matrices are represented by boldface capital letters, such as $\mathbf{W}$. The $i$-th entry of a vector $\mathbf{w}$ is denoted by $w_i$, and element $(i, j)$ of a matrix $\mathbf{W}$ is denoted by $W_{i,j}$. The $i$-th row and $j$-th column of $\mathbf{W}$ are denoted by $\mathbf{W}_{i,:}$ and $\mathbf{W}_{:,j}$, respectively. For a vector $\mathbf{w} \in \mathbb{R}^n$, let $\mathrm{diag}(\mathbf{w})$ be a diagonal matrix of size $n \times n$, where $\{\mathrm{diag}(\mathbf{w})\}_{i,i} = w_i$. For a matrix $\mathbf{W}$, let $|\mathbf{W}|$ be a matrix of the same size where $|\mathbf{W}|_{i,j} = |\mathbf{W}_{i,j}|$.

Let $[p]$ denote the set of integers from 1 to $p$. An *interaction*, $\mathcal{I}$, is a subset of all input features $[p]$ with $|\mathcal{I}| \geq 2$. For a vector $\mathbf{w} \in \mathbb{R}^p$ and $\mathcal{I} \subseteq [p]$, let $\mathbf{w}_{\mathcal{I}} \in \mathbb{R}^{|\mathcal{I}|}$ be the vector restricted to the dimensions specified by $\mathcal{I}$.

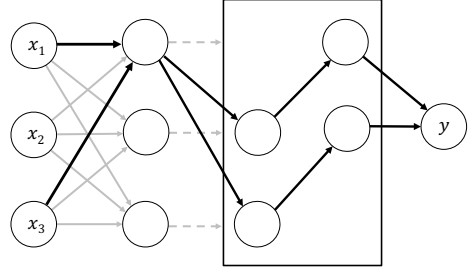

Figure 1: An illustration of an interaction within a fully connected feedforward neural network, where the box contains later layers in the network. The first hidden unit takes inputs from $x_1$ and $x_3$ with large weights and creates an interaction between them. The strength of the interaction is determined by both incoming weights and the outgoing paths between a hidden unit and the final output, $y$.

**Feedforward Neural Network**[1] Consider a feedforward neural network with $L$ hidden layers. Let $p_\ell$ be the number of hidden units in the $\ell$-th layer. We treat the input features as the 0-th layer and $p_0 = p$ is the number of input features. There are $L$ weight matrices $\mathbf{W}^{(\ell)} \in \mathbb{R}^{p_\ell \times p_{\ell-1}}$ and $L$ bias vectors $\mathbf{b}^{(\ell)} \in \mathbb{R}^{p_\ell}$, $\ell = 1, 2, \ldots, L$. Let $\phi(\cdot)$ be the activation function (nonlinearity), and let $\mathbf{w}^y \in \mathbb{R}^{p_L}$ and $b^y \in \mathbb{R}$ be the coefficients and bias for the final output. Then, the hidden units $\mathbf{h}^{(\ell)}$ of the neural network and the output $y$ with input $\mathbf{x} \in \mathbb{R}^p$ can be expressed as:

$$\mathbf{h}^{(0)} = \mathbf{x}, \quad y = (\mathbf{w}^y)^\top \mathbf{h}^{(L)} + b^y, \quad \text{and } \mathbf{h}^{(\ell)} = \phi\left(\mathbf{W}^{(\ell)} \mathbf{h}^{(\ell-1)} + \mathbf{b}^{(\ell)}\right), \quad \forall \ell = 1, 2, \ldots, L.$$

We can construct a directed acyclic graph $G = (V, E)$ based on non-zero weights, where we create vertices for input features and hidden units in the neural network and edges based on the non-zero entries in the weight matrices. See Appendix A for a formal definition.

## 3 FEATURE INTERACTIONS IN NEURAL NETWORKS

A statistical interaction describes a situation in which the joint influence of multiple variables on an output variable is not additive (Dodge, 2006; Sorokina et al., 2008). Let $x_i, i \in [p]$ be the features and $y$ be the response variable, a statistical interaction $\mathcal{I} \subseteq [p]$ exists if and only if $\mathbb{E}[y|\mathbf{x}]$, which is a function of $\mathbf{x} = (x_1, x_2, \ldots, x_p)$, contains a non-additive interaction between variables $\mathbf{x}_{\mathcal{I}}$:

---

[1] In this paper, we mainly focus on the *multilayer perceptron* architecture with ReLU activation functions, while some of our results can be generalized to a broader class of feedforward neural networks.

**Definition 1** (Non-additive Interaction). *Consider a function $f(\cdot)$ with input variables $x_i, i \in [p]$, and an interaction $\mathcal{I} \subseteq [p]$. Then $\mathcal{I}$ is a non-additive interaction of function $f(\cdot)$ if and only if there does not exist a set of functions $f_i(\cdot), \forall i \in \mathcal{I}$ where $f_i(\cdot)$ is not a function of $x_i$, such that*

$$f(\mathbf{x}) = \sum_{i \in \mathcal{I}} f_i\left(\mathbf{x}_{[p] \setminus \{i\}}\right).$$

For example, in $x_1 x_2 + \sin(x_2 + x_3 + x_4)$, there is a pairwise interaction $\{1, 2\}$ and a 3-way *higher-order* interaction $\{2, 3, 4\}$, where higher-order denotes $|\mathcal{I}| \geq 3$. Note that from the definition of statistical interaction, a *$d$-way interaction can only exist if all its corresponding $(d-1)$-interactions exist* (Sorokina et al., 2008). For example, the interaction $\{1, 2, 3\}$ can only exist if interactions $\{1, 2\}$, $\{1, 3\}$, and $\{2, 3\}$ also exist. We will often use this property in this paper.

In feedforward neural networks, statistical interactions between features, or feature interactions for brevity, are created at hidden units with nonlinear activation functions, and the influences of the interactions are propagated layer-by-layer to the final output (see Figure 1). In this section, we propose a framework to identify and quantify interactions at a hidden unit for efficient interaction detection, then the interactions are combined across hidden units in Section 4.

## 3.1 Feature Interactions at Individual Hidden Units

In feedforward neural networks, any interacting features must follow strongly weighted connections to a common hidden unit before the final output. That is, in the corresponding directed graph, interacting features will share at least one common descendant. The key observation is that non-overlapping paths in the network are aggregated via weighted summation at the final output without creating any interactions between features. The statement is rigorized in the following proposition and a proof is provided in Appendix A. The reverse of this statement, that a common descendant will create an interaction among input features, holds true in most cases.

**Proposition 2** (Interactions at Common Hidden Units). *Consider a feedforward neural network with input feature $x_i, i \in [p]$, where $y = \varphi(x_1, \ldots, x_p)$. For any interaction $\mathcal{I} \subset [p]$ in $\varphi(\cdot)$, there exists a vertex $v_{\mathcal{I}}$ in the associated directed graph such that $\mathcal{I}$ is a subset of the ancestors of $v_{\mathcal{I}}$ at the input layer (i.e., $\ell = 0$).*

In general, the weights in a neural network are nonzero, in which case Proposition 2 blindly infers that all features are interacting. For example, in a neural network with just a single hidden layer, any hidden unit in the network can imply up to $2^{\|\mathbf{W}_{j,:}\|_0}$ potential interactions, where $\|\mathbf{W}_{j,:}\|_0$ is the number of nonzero values in the weight vector $\mathbf{W}_{j,:}$ for the $j$-th hidden unit. Managing the large solution space of interactions based on nonzero weights requires us to characterize the relative importance of interactions, so we must mathematically define the concept of interaction strength. In addition, we limit the search complexity of our task by only quantifying interactions created at the first hidden layer, which is important for fast interaction detection and sufficient for high detection performance based on empirical evaluation (see evaluation in Section 5.2 and Table 2).

Consider a hidden unit in the first layer: $\phi\left(\mathbf{w}^\top \mathbf{x} + b\right)$, where $\mathbf{w}$ is the associated weight vector and $\mathbf{x}$ is the input vector. While having the weight $w_i$ of each feature $i$, the correct way of summarizing feature weights for defining interaction strength is not trivial. For an interaction $\mathcal{I} \subseteq [p]$, we propose to use an average of the relevant feature weights $\mathbf{w}_{\mathcal{I}}$ as the surrogate for the interaction strength: $\mu(|\mathbf{w}_{\mathcal{I}}|)$, where $\mu(\cdot)$ is the averaging function for an interaction that represents the interaction strength due to feature weights.

We provide guidance on how $\mu$ should be defined by first considering representative averaging functions from the generalized mean family: maximum value, root mean square, arithmetic mean, geometric mean, harmonic mean, and minimum value (Bullen et al., 1988). These options can be narrowed down by accounting for intuitive properties of interaction strength : 1) interaction strength is evaluated as zero whenever an interaction does not exist (one of the features has zero weight); 2) interaction strength does not decrease with any increase in magnitude of feature weights; 3) interaction strength is less sensitive to changes in large feature weights.

While the first two properties place natural constraints on interaction strength behavior, the third property is subtle in its intuition. Consider the scaling between the magnitudes of multiple feature weights, where one weight has much higher magnitude than the others. In the worst case, there is

one large weight in magnitude while the rest are near zero. If the large weight grows in magnitude, then interaction strength may not change significantly, but if instead the smaller weights grow at the same rate, then interaction strength should strictly increase. As a result, maximum value, root mean square, and arithmetic mean should be ruled out because they do not satisfy either property 1 or 3.

## 3.2 MEASURING THE INFLUENCE OF HIDDEN UNITS

Our definition of interaction strength at individual hidden units is not complete without considering their outgoing paths, because an outgoing path of zero weight cannot contribute an interaction to the final output. To propose a way of quantifying the influence of an outgoing path on the final output, we draw inspiration from Garson's algorithm (Garson, 1991; Goh, 1995), which instead of computing the influence of a hidden unit, computes the influence of features on the output. This is achieved by cumulative matrix multiplications of the absolute values of weight matrices. In the following, we propose our definition of hidden unit influence, then prove that this definition upper bounds the gradient magnitude of the hidden unit with its activation function. To represent the influence of a hidden unit $i$ at the $\ell$-th hidden layer, we define the *aggregated weight* $z_i^{(\ell)}$,

$$\mathbf{z}^{(\ell)} = |\mathbf{w}^y|^\top \left|\mathbf{W}^{(L)}\right| \cdot \left|\mathbf{W}^{(L-1)}\right| \cdots \left|\mathbf{W}^{(\ell+1)}\right|,$$

where $\mathbf{z}^{(\ell)} \in \mathbb{R}^{p_\ell}$. This definition upper bounds the gradient magnitudes of hidden units because it computes Lipschitz constants for the corresponding units. Gradients have been commonly used as variable importance measures in neural networks, especially input gradients which compute directions normal to decision boundaries (Ross et al., 2017; Goodfellow et al., 2015; Simonyan et al., 2013). Thus, an upper bound on the gradient magnitude approximates how important the variable can be. A full proof is shown in Appendix C.

**Lemma 3** (Neural Network Lipschitz Estimation). *Let the activation function $\phi\left(\cdot\right)$ be a 1-Lipschitz function. Then the output $y$ is $z_i^{(\ell)}$-Lipschitz with respect to $h_i^{(\ell)}$.*

## 3.3 QUANTIFYING INTERACTION STRENGTH

We now combine our definitions from Sections 3.1 and 3.2 to obtain the interaction strength $\omega_i(\mathcal{I})$ of a potential interaction $\mathcal{I}$ at the $i$-th unit in the first hidden layer $h_i^{(1)}$,

$$\omega_i(\mathcal{I}) = z_i^{(1)} \mu\left(\left|\mathbf{W}_{i,\mathcal{I}}^{(1)}\right|\right). \tag{1}$$

Note that $\omega_i(\mathcal{I})$ is defined on a single hidden unit, and it is agnostic to scaling ambiguity within a ReLU based neural network. In Section 4, we discuss our scheme of aggregating strengths across hidden units, so we can compare interactions of different orders.

# 4 INTERACTION DETECTION

In this section, we propose our feature interaction detection algorithm NID, which can extract interactions of all orders without individually testing for each of them. Our methodology for interaction detection is comprised of three main steps: 1) train a feedforward network with regularization, 2) interpret learned weights to obtain a ranking of interaction candidates, and 3) determine a cutoff for the top-$K$ interactions.

## 4.1 ARCHITECTURE

Data often contains both statistical interactions and main effects (Winer et al., 1971). Main effects describe the univariate influences of variables on an outcome variable. We study two architectures: *MLP* and *MLP-M*. *MLP* is a standard multilayer perceptron, and *MLP-M* is an *MLP* with additional univariate networks summed at the output (Figure 2). The univariate networks are intended to discourage the modeling of main effects away from the standard *MLP*, which can create spurious interactions using the main effects. When training the neural networks, we apply sparsity regularization on the *MLP* portions of the architectures to 1) suppress unimportant interacting paths and 2) push the modeling of main effects towards any univariate networks. We note that this approach can also generalize beyond sparsity regularization (Appendix G).

---

**Algorithm 1** `NID` Greedy Ranking Algorithm

---

**Input:** input-to-first hidden layer weights $\mathbf{W}^{(1)}$, aggregated weights $\mathbf{z}^{(1)}$
**Output:** ranked list of interaction candidates $\{\mathcal{I}_i\}_{i=1}^m$
 1: $d \leftarrow$ initialize an empty dictionary mapping interaction candidate to interaction strength
 2: **for** each row $\mathbf{w}'$ of $\mathbf{W}^{(1)}$ indexed by $r$ **do**
 3:     **for** $j = 2$ to $p$ **do**
 4:         $\mathcal{I} \leftarrow$ sorted indices of top $j$ weights in $\mathbf{w}'$
 5:         $d[\mathcal{I}] \leftarrow d[\mathcal{I}] + z_r^{(1)}\mu\left(|\mathbf{w}'_{\mathcal{I}}|\right)$
 6: $\{\mathcal{I}_i\}_{i=1}^m \leftarrow$ interaction candidates in $d$ sorted by their strengths in descending order

---

## 4.2 Ranking Interactions of Variable Order

We design a greedy algorithm that generates a ranking of interaction candidates by only considering, at each hidden unit, the top-ranked interactions of every order, where $2 \leq |\mathcal{I}| \leq p$, thereby drastically reducing the search space of potential interactions while still considering all orders. The greedy algorithm (Algorithm 1) traverses the learned input weight matrix $\mathbf{W}^{(1)}$ across hidden units and selects only top-ranked interaction candidates per hidden unit based on their interaction strengths (Equation 1). By selecting the top-ranked interactions of every order and summing their respective strengths across hidden units, we obtain final interaction strengths, allowing variable-order interaction candidates to be ranked relative to each other. For this algorithm, we set the averaging function $\mu(\cdot) = \min(\cdot)$ based on its performance in experimental evaluation (Section 5.1).

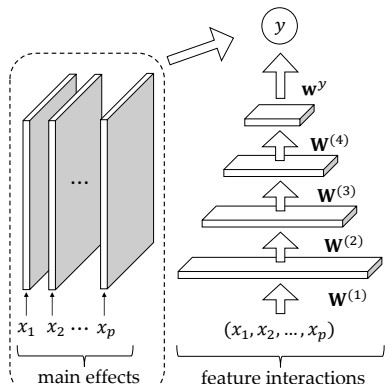

Figure 2: Standard feedforward neural network for interaction detection, with optional univariate networks

With the averaging function set to $\min(\cdot)$, Algorithm 1's greedy strategy automatically improves the ranking of a higher-order interaction over its redundant subsets[2] (for redundancy, see Definition 1). This allows the higher-order interaction to have a better chance of ranking above any false positives and being captured in the cutoff stage. We justify this improvement by proving Theorem 4 under a mild assumption.

**Theorem 4** (Improving the ranking of higher-order interactions). *Let $\mathcal{R}$ be the set of interactions proposed by Algorithm 1 with $\mu(\cdot) = \min(\cdot)$, let $\mathcal{I} \in \mathcal{R}$ be a $d$-way interaction where $d \geq 3$, and let $\mathcal{S}$ be the set of subset $(d-1)$-way interactions of $\mathcal{I}$ where $|\mathcal{S}| = d$. Assume that for any hidden unit $j$ which proposed $s \in \mathcal{S} \cap \mathcal{R}$, $\mathcal{I}$ will also be proposed at the same hidden unit, and $\omega_j(\mathcal{I}) > \frac{1}{d}\omega_j(s)$. Then, one of the following must be true: a) $\exists s \in \mathcal{S} \cap \mathcal{R}$ ranked lower than $\mathcal{I}$, i.e., $\omega(\mathcal{I}) > \omega(s)$, or b) $\exists s \in \mathcal{S}$ where $s \notin \mathcal{R}$.*

The full proof is included in Appendix D. Under the noted assumption, the theorem in part a) shows that a $d$-way interaction will improve over one its $d-1$ subsets in rankings as long as there is no sudden drop from the weight of the $(d-1)$-way to the $d$-way interaction at the same hidden units. We note that the improvement extends to b) as well, when $d = |\mathcal{S} \cap \mathcal{R}| > 1$.

Lastly, we note that Algorithm 1 assumes there are at least as many first-layer hidden units as there are the true number of non-redundant interactions. In practice, we use an arbitrarily large number of first-layer hidden units because true interactions are initially unknown.

## 4.3 Cutoff on Interaction Ranking

In order to predict the true top-$K$ interactions $\{\mathcal{I}_i\}_{i=1}^K$, we must find a cutoff point on our interaction ranking from Section 4.2. We obtain this cutoff by constructing a Generalized Additive Model (*GAM*) with interactions:

---

[2]When a higher-order interaction is ranked above any of its subset interactions, those subset interactions can be automatically pruned from the ranking due to their redundancy.

Table 1: Test suite of data-generating functions

| | |
|---|---|
| $F_1(\mathbf{x})$ | $\pi^{x_1 x_2}\sqrt{2x_3} - \sin^{-1}(x_4) + \log(x_3 + x_5) - \frac{x_9}{x_{10}}\sqrt{\frac{x_7}{x_8}} - x_2 x_7$ |
| $F_2(\mathbf{x})$ | $\pi^{x_1 x_2}\sqrt{2|x_3|} - \sin^{-1}(0.5x_4) + \log(|x_3 + x_5| + 1) + \frac{x_9}{1 + |x_{10}|}\sqrt{\frac{x_7}{1 + |x_8|}} - x_2 x_7$ |
| $F_3(\mathbf{x})$ | $\exp|x_1 - x_2| + |x_2 x_3| - x_3^{2|x_4|} + \log(x_4^2 + x_5^2 + x_7^2 + x_8^2) + x_9 + \frac{1}{1 + x_{10}^2}$ |
| $F_4(\mathbf{x})$ | $\exp|x_1 - x_2| + |x_2 x_3| - x_3^{2|x_4|} + (x_1 x_4)^2 + \log(x_4^2 + x_5^2 + x_7^2 + x_8^2) + x_9 + \frac{1}{1 + x_{10}^2}$ |
| $F_5(\mathbf{x})$ | $\frac{1}{1 + x_1^2 + x_2^2 + x_3^2} + \sqrt{\exp(x_4 + x_5)} + |x_6 + x_7| + x_8 x_9 x_{10}$ |
| $F_6(\mathbf{x})$ | $\exp(|x_1 x_2| + 1) - \exp(|x_3 + x_4| + 1) + \cos(x_5 + x_6 - x_8) + \sqrt{x_8^2 + x_9^2 + x_{10}^2}$ |
| $F_7(\mathbf{x})$ | $(\arctan(x_1) + \arctan(x_2))^2 + \max(x_3 x_4 + x_6, 0) - \frac{1}{1 + (x_4 x_5 x_6 x_7 x_8)^2} + \left(\frac{|x_7|}{1 + |x_9|}\right)^5 + \sum_{i=1}^{10} x_i$ |
| $F_8(\mathbf{x})$ | $x_1 x_2 + 2^{x_3 + x_5 + x_6} + 2^{x_3 + x_4 + x_5 + x_7} + \sin(x_7 \sin(x_8 + x_9)) + \arccos(0.9x_{10})$ |
| $F_9(\mathbf{x})$ | $\tanh(x_1 x_2 + x_3 x_4)\sqrt{|x_5|} + \exp(x_5 + x_6) + \log((x_6 x_7 x_8)^2 + 1) + x_9 x_{10} + \frac{1}{1 + |x_{10}|}$ |
| $F_{10}(\mathbf{x})$ | $\sinh(x_1 + x_2) + \arccos(\tanh(x_3 + x_5 + x_7)) + \cos(x_4 + x_5) + \sec(x_7 x_9)$ |

$$c_K(\mathbf{x}) = \sum_{i=1}^{p} g_i(x_i) + \sum_{i=1}^{K} g_i'(\mathbf{x}_{\mathcal{I}}),$$

where $g_i(\cdot)$ captures the main effects, $g_i'(\cdot)$ captures the interactions, and both $g_i$ and $g_i'$ are small feedforward networks trained jointly via backpropagation. We refer to this model as *MLP-Cutoff*.

We gradually add top-ranked interactions to the *GAM*, increasing $K$, until *GAM* performance on a validation set plateaus. The exact plateau point can be found by early stopping or other heuristic means, and we report $\{\mathcal{I}_i\}_{i=1}^{K}$ as the identified feature interactions.

### 4.4 PAIRWISE INTERACTION DETECTION

A variant to our interaction ranking algorithm tests for all pairwise interactions. Pairwise interaction detection has been a standard problem in the interaction detection literature (Lou et al., 2013; Fan et al., 2016) due to its simplicity. Modeling pairwise interactions is also the *de facto* objective of many successful machine learning algorithms such as factorization machines (Rendle, 2010) and hierarchical lasso (Bien et al., 2013).

We rank all pairs of features $\{i, j\}$ according to their interaction strengths $\omega(\{i, j\})$ calculated on the first hidden layer, where again the averaging function is $\min(\cdot)$, and $\omega(\{i, j\}) = \sum_{s=1}^{p_1} \omega_s(\{i, j\})$. The higher the rank, the more likely the interaction exists.

## 5 EXPERIMENTS

In this section, we discuss our experiments on both simulated and real-world datasets to study the performance of our approach on interaction detection.

### 5.1 EXPERIMENTAL SETUP

**Averaging Function** Our proposed `NID` framework relies on the selection of an averaging function (Sections 3.1, 4.2, and 4.4). We experimentally determined the averaging function by comparing representative functions from the generalized mean family (Bullen et al., 1988): maximum, root mean square, arithmetic mean, geometric mean, harmonic mean, and minimum, intuitions behind which were discussed in Section 3.1. To make the comparison, we used a test suite of 10 synthetic functions, which consist of a variety of interactions of varying order and overlap, as shown in Table 1. We trained 10 trials of *MLP* and *MLP-M* on each of the synthetic functions, obtained interaction rankings with our proposed greedy ranking algorithm (Algorithm 1), and counted the total number of correct interactions ranked before any false positive. In this evaluation, we ignore predicted interactions that are subsets of true higher-order interactions because the subset interactions are redundant (Section 2). As seen in Figure 3, the number of true top interactions we recover is highest

with the averaging function, *minimum*, which we will use in all of our experiments. A simple analytical study on a bivariate hidden unit also suggests that the minimum is closely correlated with interaction strength (Appendix B).

**Neural Network Configuration** We trained feedforward networks of *MLP* and *MLP-M* architectures to obtain interaction rankings, and we trained *MLP-Cutoff* to find cutoffs on the rankings. In our experiments, all networks that model feature interactions consisted of four hidden layers with first-to-last layer sizes of: 140, 100, 60, and 20 units. In contrast, all individual univariate networks had three hidden layers with sizes of: 10, 10, and 10 units. All networks used ReLU activation and were trained using backpropagation. In the cases of *MLP-M* and *MLP-Cutoff*, summed networks were trained jointly. The objective functions were mean-squared error for regression and cross-entropy for classification tasks. On the synthetic test suite, *MLP* and *MLP-M* were trained with L1 constants in the range of 5e-6 to 5e-4, based on parameter tuning on a validation set. On real-world datasets, L1 was fixed at 5e-5. *MLP-Cutoff* used a fixed L2 constant of 1e-4 in all experiments involving cutoff. Early stopping was used to prevent overfitting.

**Datasets** We study our interaction detection framework on both simulated and real-world experiments. For simulated experiments, we used a test suite of synthetic functions, as shown in Table 1. The test functions were designed to have

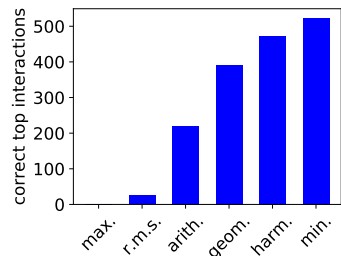

Figure 3: A comparison of averaging functions by the total number of correct interactions ranked before any false positives, evaluated on the test suite (Table 1) over 10 trials. $x$-axis labels are maximum, root mean square, arithmetic mean, geometric mean, harmonic mean, and minimum.

a mixture of pairwise and higher-order interactions, with varying order, strength, nonlinearity, and overlap. $F_1$ is a commonly used function in interaction detection literature (Hooker, 2004; Sorokina et al., 2008; Lou et al., 2013). All features were uniformly distributed between $-1$ and 1 except in $F_1$, where we used the same variable ranges as reported in literature (Hooker, 2004). In all synthetic experiments, we used random train/valid/test splits of $1/3$ each on 30k data points.

We use four real-world datasets, of which two are regression datasets, and the other two are binary classification datasets. The datasets are a mixture of common prediction tasks in the cal housing and bike sharing datasets, a scientific discovery task in the higgs boson dataset, and an example of very-high order interaction detection in the letter dataset. Specifically, the cal housing dataset is a regression dataset with 21k data points for predicting California housing prices (Pace & Barry, 1997). The bike sharing dataset contains 17k data points of weather and seasonal information to predict the hourly count of rental bikes in a bikeshare system (Fanaee-T & Gama, 2014). The higgs boson dataset has 800k data points for classifying whether a particle environment originates from the decay of a Higgs Boson (Adam-Bourdarios et al., 2014). Lastly, the letter recognition dataset contains 20k data points of transformed features for binary classification of letters on a pixel display (Frey & Slate, 1991). For all real-world data, we used random train/valid/test splits of $80/10/10$.

**Baselines** We compare the performance of NID to that of three baseline interaction detection methods. Two-Way *ANOVA* (Wonnacott & Wonnacott, 1972) utilizes linear models to conduct significance tests on the existence of interaction terms. *Hierarchical lasso* (HierLasso) (Bien et al., 2013) applies lasso feature selection to extract pairwise interactions. *RuleFit* (Friedman & Popescu, 2008) contains a statistic to measure pairwise interaction strength using partial dependence functions. *Additive Groves* (AG) (Sorokina et al., 2008) is a nonparameteric means of testing for interactions by placing structural constraints on an additive model of regression trees. *AG* is a reference method for interaction detection because it directly detects interactions based on their non-additive definition.

## 5.2 PAIRWISE INTERACTION DETECTION

As discussed in Section 4, our framework NID can be used for pairwise interaction detection. To evaluate this approach, we used datasets generated by synthetic functions $F_1$-$F_{10}$ (Table 1) that contain a mixture of pairwise and higher-order interactions, where in the case of higher-order interactions we tested for their pairwise subsets as in Sorokina et al. (2008); Lou et al. (2013). AUC

Table 2: AUC of pairwise interaction strengths proposed by `NID` and baselines on a test suite of synthetic functions (Table 1). *ANOVA*, *HierLasso*, and *RuleFit* are deterministic.

| | ANOVA | HierLasso | RuleFit | AG | NID, *MLP* | NID, *MLP-M* |
|---|---|---|---|---|---|---|
| $F_1(\mathbf{x})$ | 0.992 | 1.00 | 0.754 | $1 \pm 0.0$ | $0.970 \pm 9.2e{-}3$ | $0.995 \pm 4.4e{-}3$ |
| $F_2(\mathbf{x})$ | 0.468 | 0.636 | 0.698 | $0.88 \pm 1.4e{-}2$ | $0.79 \pm 3.1e{-}2$ | $0.85 \pm 3.9e{-}2$ |
| $F_3(\mathbf{x})$ | 0.657 | 0.556 | 0.815 | $1 \pm 0.0$ | $0.999 \pm 2.0e{-}3$ | $1 \pm 0.0$ |
| $F_4(\mathbf{x})$ | 0.563 | 0.634 | 0.689 | $0.999 \pm 1.4e{-}2$ | $0.85 \pm 6.7e{-}2$ | $0.996 \pm 4.7e{-}3$ |
| $F_5(\mathbf{x})$ | 0.544 | 0.625 | 0.797 | $0.67 \pm 5.7e{-}2$ | $1 \pm 0.0$ | $1 \pm 0.0$ |
| $F_6(\mathbf{x})$ | 0.780 | 0.730 | 0.811 | $0.64 \pm 1.4e{-}2$ | $0.98 \pm 6.7e{-}2$ | $0.70 \pm 4.8e{-}2$ |
| $F_7(\mathbf{x})$ | 0.726 | 0.571 | 0.666 | $0.81 \pm 4.9e{-}2$ | $0.84 \pm 1.7e{-}2$ | $0.82 \pm 2.2e{-}2$ |
| $F_8(\mathbf{x})$ | 0.929 | 0.958 | 0.946 | $0.937 \pm 1.4e{-}3$ | $0.989 \pm 4.4e{-}3$ | $0.989 \pm 4.5e{-}3$ |
| $F_9(\mathbf{x})$ | 0.783 | 0.681 | 0.584 | $0.808 \pm 5.7e{-}3$ | $0.83 \pm 5.3e{-}2$ | $0.83 \pm 3.7e{-}2$ |
| $F_{10}(\mathbf{x})$ | 0.765 | 0.583 | 0.876 | $1 \pm 0.0$ | $0.995 \pm 9.5e{-}3$ | $0.99 \pm 2.1e{-}2$ |
| average | 0.721 | 0.698 | 0.764 | $0.87 \pm 1.4e{-}2$ | $\mathbf{0.92^*} \pm 2.3e{-}2$ | $\mathbf{0.92} \pm 1.8e{-}2$ |

*Note: The high average AUC of `NID`, *MLP* is heavily influenced by $F_6$.

scores of interaction strength proposed by baseline methods and `NID` for both *MLP* and *MLP-M* are shown in Table 2. We ran ten trials of *AG* and `NID` on each dataset and removed two trials with highest and lowest AUC scores.

When comparing the AUCs of `NID` applied to *MLP* and *MLP-M*, we observe that the scores of *MLP-M* tend to be comparable or better, except the AUC for $F_6$. On one hand, *MLP-M* performed better on $F_2$ and $F_4$ because these functions contain main effects that *MLP* would model as spurious interactions with other variables. On the other hand, *MLP-M* performed worse on $F_6$ because it modeled *spurious main effects* in the $\{8, 9, 10\}$ interaction. Specifically, $\{8, 9, 10\}$ can be approximated as independent parabolas for each variable (shown in Appendix I). In our analyses of `NID`, we mostly focus on *MLP-M* because handling main effects is widely considered an important problem in interaction detection (Bien et al., 2013; Lim & Hastie, 2015; Kong et al., 2017). Comparing the AUCs of *AG* and `NID` for *MLP-M*, the scores tend to close, except for $F_5$, $F_6$, and $F_8$, where `NID` performs significantly better than *AG*. This performance difference may be due to limitations on the model capacity of *AG*, which is tree-based. In comparison to *ANOVA*, *HierLasso* and *RuleFit*, `NID`-*MLP-M* generally performs on par or better. This is expected for *ANOVA* and *HierLasso* because they are based on quadratic models, which can have difficulty approximating the interaction nonlinearities present in the test suite.

In Figure 4, heat maps of synthetic functions show the relative strengths of all possible pairwise interactions as interpreted from *MLP-M*, and ground truth is indicated by red cross-marks. The interaction strengths shown are normally high at the cross-marks. An exception is $F_6$, where `NID` proposes weak or negligible interaction strengths at the cross-marks corresponding to the $\{8, 9, 10\}$ interaction, which is consistent with previous remarks about this interaction. Besides $F_6$, $F_7$ also shows erroneous interaction strengths; however, comparative detection performance by the baselines is similarly poor. Interaction strengths are also visualized on real-world datasets via heat maps (Figure 5). For example, in the cal housing dataset, there is a high-strength interaction between $x_1$ and $x_2$. These variables mean longitude and latitude respectively, and it is clear to see that the outcome variable, California housing price, should indeed strongly depend on geographical location. We further observe high-strength interactions appearing in the heat maps of the bike sharing, higgs boson dataset, and letter datasets. For example, all feature pairs appear to be interacting in the letter dataset. The binary classification task from the letter dataset is to distinguish letters A-M from N-Z using 16 pixel display features. Since the decision boundary between A-M and N-Z is not obvious, it would make sense that a neural network learns a highly interacting function to make the distinction.

### 5.3 HIGHER-ORDER INTERACTION DETECTION

We use our greedy interaction ranking algorithm (Algorithm 1) to perform higher-order interaction detection without an exponential search of interaction candidates. We first visualize our higher-order interaction detection algorithm on synthetic and real-world datasets, then we show how the predictive capability of detected interactions closes the performance gap between *MLP-Cutoff* and *MLP-M*. Next, we discuss our experiments comparing `NID` and *AG* with added noise, and lastly we verify that our algorithm obtains significant improvements in runtime.

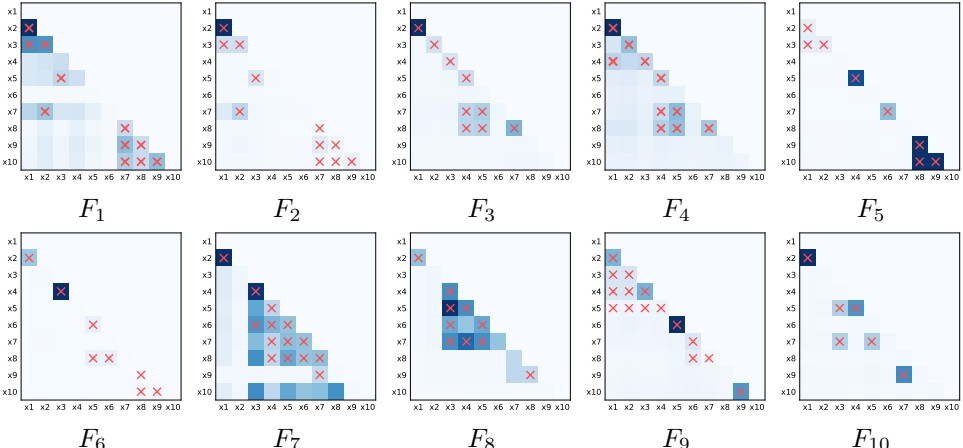

Figure 4: Heat maps of pairwise interaction strengths proposed by our NID framework on *MLP-M* for datasets generated by functions $F_1$-$F_{10}$ (Table 1). Cross-marks indicate ground truth interactions.

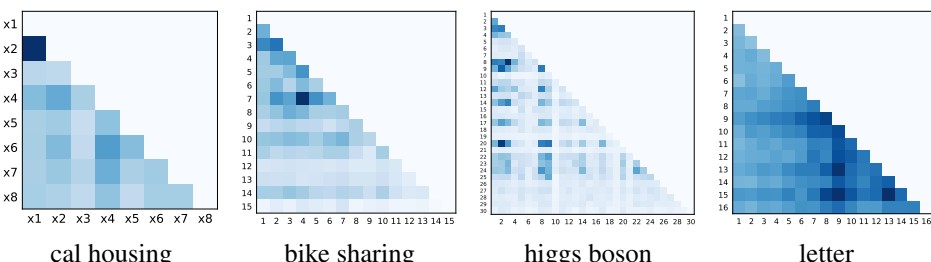

Figure 5: Heat maps of pairwise interaction strengths proposed by our NID framework on *MLP-M* for real-world datasets.

We visualize higher-order interaction detection on synthetic and real-world datasets in Figures 6 and 7 respectively. The plots correspond to higher-order interaction detection as the ranking cutoff is applied (Section 4.3). The interaction rankings generated by NID for *MLP-M* are shown on the $x$-axes, and the blue bars correspond to the validation performance of *MLP-Cutoff* as interactions are added. For example, the plot for cal housing shows that adding the first interaction significantly reduces RMSE. We keep adding interactions into the model until reaching a cutoff point. In our experiments, we use a cutoff heuristic where interactions are no longer added after *MLP-Cutoff*'s validation performance reaches or surpasses *MLP-M*'s validation performance (represented by horizontal dashed lines).

As seen with the red cross-marks, our method finds true interactions in the synthetic data of $F_1$-$F_{10}$ before the cutoff point. Challenges with detecting interactions are again mainly associated with $F_6$ and $F_7$, which have also been difficult for baselines in the pairwise detection setting (Table 2). For the cal housing dataset, we obtain the top interaction $\{1, 2\}$ just like in our pairwise test (Figure 5, cal housing), where now the $\{1, 2\}$ interaction contributes a significant improvement in *MLP-Cutoff* performance. Similarly, from the letter dataset we obtain a 16-way interaction, which is consistent with its highly interacting pairwise heat map (Figure 5, letter). For the bike sharing and higgs boson datasets, we note that even when considering many interactions, *MLP-Cutoff* eventually reaches the cutoff point with a relatively small number of superset interactions. This is because many subset interactions become redundant when their corresponding supersets are found.

In our evaluation of interaction detection on real-world data, we study detected interactions via their predictive performance. By comparing the test performance of *MLP-Cutoff* and *MLP-M* with respect to *MLP-Cutoff* without any interactions (*MLP-Cutoff*$^\varnothing$), we can compute the relative test performance improvement obtained by including detected interactions. These relative performance improvements are shown in Table 3 for the real-world datasets as well as four selected synthetic datasets, where performance is averaged over ten trials per dataset. The results of this study show

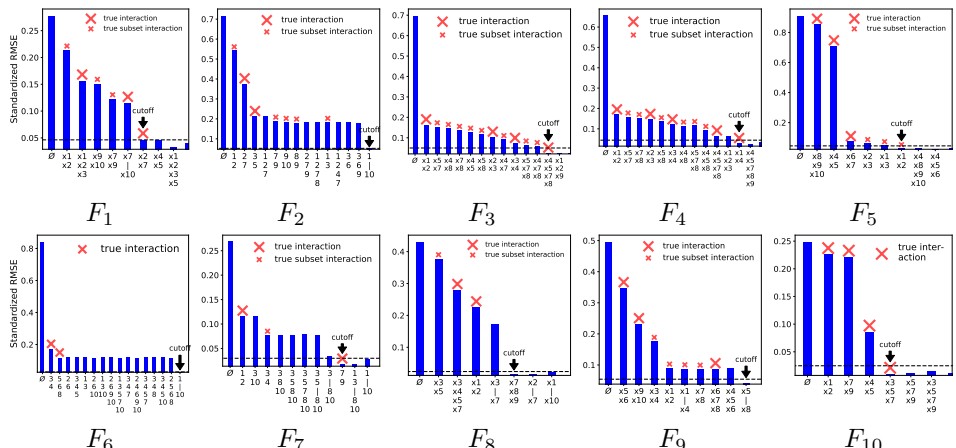

Figure 6: *MLP-Cutoff* error with added top-ranked interactions (along $x$-axis) of $F_1$-$F_{10}$ (Table 1), where the interaction rankings were generated by the `NID` framework applied to *MLP-M*. Red cross-marks indicate ground truth interactions, and Ø denotes *MLP-Cutoff* without any interactions. Subset interactions become redundant when their true superset interactions are found.

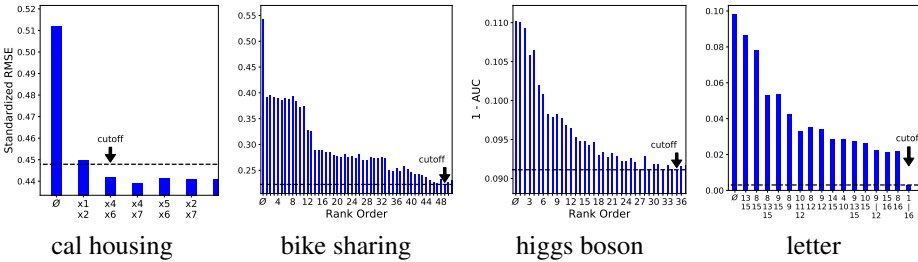

Figure 7: *MLP-Cutoff* error with added top-ranked interactions (along $x$-axis) of real-world datasets (Table 1), where the interaction rankings were generated by the `NID` framework on *MLP-M*. Ø denotes *MLP-Cutoff* without any interactions.

that a relatively small number of interactions of variable order are highly predictive of their corresponding datasets, as true interactions should.

We further study higher-order interaction detection of our `NID` framework by comparing it to *AG* in both interaction ranking quality and runtime. To assess ranking quality, we design a metric, *top-rank recall*, which computes a recall of proposed interaction rankings by only considering those interactions that are correctly ranked before any false positive. The number of top correctly-ranked interactions is then divided by the true number of interactions. Because subset interactions are redundant in the presence of corresponding superset interactions, only such superset interactions can count as true interactions, and our metric ignores any subset interactions in the ranking. We compute the *top-rank recall* of `NID` on *MLP* and *MLP-M*, the scores of which are averaged across all tests in the test suite of synthetic functions (Table 1) with 10 trials per test function. For each test, we remove two trials with max and min recall. We conduct the same tests using the state-of-the-art interaction detection method *AG*, except with only one trial per test because *AG* is very computationally expensive to run. In Figure 8a, we show *top-rank recall* of `NID` and *AG* at different Gaussian noise levels[3], and in Figure 8b, we show runtime comparisons on real-world and synthetic datasets. As shown, `NID` can obtain similar top-rank recall as *AG* while running orders of magnitude times faster.

## 5.4 LIMITATIONS

In higher-order interaction detection, our `NID` framework can have difficulty detecting interactions from functions with interlinked interacting variables. For example, a clique $x_1x_2 + x_1x_3 + x_2x_3$ only

---

[3]Gaussian noise was to applied to both features and the outcome variable after standard scaling all variables.

Table 3: Test performance improvement when adding top-$K$ discovered interactions to *MLP-Cutoff* on real-world datasets and select synthetic datasets. Here, the median $\bar{K}$ excludes subset interactions, and $|\bar{\mathcal{I}}|$ denotes average interaction cardinality. RMSE values are standard scaled.

| Dataset | $p$ | Relative Performance Improvement | Absolute Performance Improvement | $\bar{K}$ | $|\bar{\mathcal{I}}|$ |
|---|---|---|---|---|---|
| cal housing | 8 | $99\% \pm 4.0\%$ | $0.09 \pm 1.3\mathrm{e}{-2}$ RMSE | 2 | 2.0 |
| bike sharing | 12 | $98.8\% \pm 0.89\%$ | $0.331 \pm 4.6\mathrm{e}{-3}$ RMSE | 12 | 4.7 |
| higgs boson | 30 | $98\% \pm 1.4\%$ | $0.0188 \pm 5.9\mathrm{e}{-4}$ AUC | 11 | 4.0 |
| letter | 16 | $101.1\% \pm 0.58\%$ | $0.103 \pm 5.8\mathrm{e}{-3}$ AUC | 1 | 16 |
| $F_3(\mathbf{x})$ | 10 | $104.1\% \pm 0.21\%$ | $0.672 \pm 2.2\mathrm{e}{-3}$ RMSE | 4 | 2.5 |
| $F_5(\mathbf{x})$ | 10 | $102.0\% \pm 0.30\%$ | $0.875 \pm 2.2\mathrm{e}{-3}$ RMSE | 6 | 2.2 |
| $F_7(\mathbf{x})$ | 10 | $105.2\% \pm 0.30\%$ | $0.2491 \pm 6.4\mathrm{e}{-4}$ RMSE | 3 | 3.7 |
| $F_{10}(\mathbf{x})$ | 10 | $105.5\% \pm 0.50\%$ | $0.234 \pm 1.5\mathrm{e}{-3}$ RMSE | 4 | 2.3 |

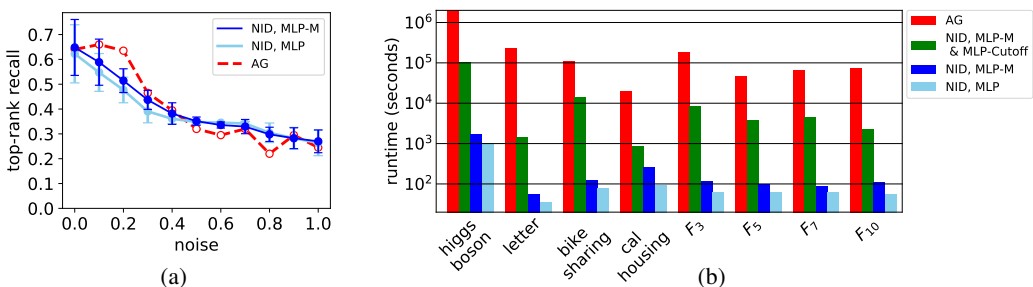

(a)  (b)

Figure 8: Comparisons between *AG* and `NID` in higher-order interaction detection. (a) Comparison of top-ranked recall at different noise levels on the synthetic test suite (Table 1), (b) comparison of runtimes, where `NID` runtime with and without cutoff are both measured. `NID` detects interactions with top-rank recall close to the state-of-the-art *AG* while running orders of magnitude times faster.

contains pairwise interactions. When detecting pairwise interactions (Section 5.2), `NID` often obtains an AUC of 1. However, in higher-order interaction detection, the interlinked pairwise interactions are often confused for single higher-order interactions. This issue could mean that our higher-order interaction detection algorithm fails to separate interlinked pairwise interactions encoded in a neural network, or the network approximates interlinked low-order interactions as higher-order interactions. Another limitation of our framework is that it sometimes detects spurious interactions or misses interactions as a result of correlations between features; however, correlations are known to cause such problems for any interaction detection method (Sorokina et al., 2008; Lou et al., 2013).

## 6 CONCLUSION

We presented our `NID` framework, which detects statistical interactions by interpreting the learned weights of a feedforward neural network. The framework has the practical utility of accurately detecting general types of interactions without searching an exponential solution space of interaction candidates. Our core insight was that interactions between features must be modeled at common hidden units, and our framework decoded the weights according to this insight.

In future work, we plan to detect feature interactions by accounting for common units in intermediate hidden layers of feedforward networks. We would also like to use the perspective of interaction detection to interpret weights in other deep neural architectures.

### ACKNOWLEDGMENTS

We are very thankful to Xinran He, Natali Ruchansky, Meisam Razaviyayn, Pavankumar Murali (IBM), Harini Eavani (Intel), and anonymous reviewers for their thoughtful advice. This work was supported by National Science Foundation awards IIS-1254206, IIS-1539608 and USC Annenberg Fellowships for MT and DC.

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

# A   PROOF AND DISCUSSION FOR PROPOSITION 2

Given a trained feedforward neural network as defined in Section 2.3, we can construct a directed acyclic graph $G = (V, E)$ based on non-zero weights as follows. We create a vertex for each input feature and hidden unit in the neural network: $V = \{v_{\ell,i} | \forall i, \ell\}$, where $v_{\ell,i}$ is the vertex corresponding to the $i$-th hidden unit in the $\ell$-th layer. Note that the final output $y$ is not included. We create edges based on the non-zero entries in the weight matrices, i.e., $E = \{(v_{\ell-1,i}, v_{\ell,j}) | \mathbf{W}^{\ell}_{j,i} \neq 0, \forall i, j, \ell\}$. Note that under the graph representation, the value of any hidden unit is a function of parent hidden units. In the following proposition, we will use vertices and hidden units interchangeably.

**Proposition 2** (Interactions at Common Hidden Units). *Consider a feedforward neural network with input feature $x_i, i \in [p]$, where $y = \varphi(x_1, \ldots, x_p)$. For any interaction $\mathcal{I} \subset [p]$ in $\varphi(\cdot)$, there exists a vertex $v_{\mathcal{I}}$ in the associated directed graph such that $\mathcal{I}$ is a subset of the ancestors of $v_{\mathcal{I}}$ at the input layer (i.e., $\ell = 0$).*

*Proof.* We prove Proposition 2 by contradiction.

Let $\mathcal{I}$ be an interaction where there is no vertex in the associated graph which satisfies the condition. Then, for any vertex $v_{L,i}$ at the $L$-th layer, the value $f_i$ of the corresponding hidden unit is a function of its ancestors at the input layer $\mathcal{I}_i$ where $\mathcal{I} \not\subset \mathcal{I}_i$.

Next, we group the hidden units at the $L$-th layer into non-overlapping subsets by the first missing feature with respect to the interaction $\mathcal{I}$. That is, for element $i$ in $\mathcal{I}$, we create an index set $\mathcal{S}_i \in [p_L]$:

$$\mathcal{S}_i = \{j \in [p_L] | i \notin \mathcal{I}_j \text{ and } \forall i' < i, j \notin \mathcal{S}_{i'}\}.$$

Note that the final output of the network is a weighed summation over the hidden units at the $L$-th layer:

$$\varphi(\mathbf{x}) = b^y + \sum_{i \in \mathcal{I}} \sum_{j \in \mathcal{S}_i} w^y_j f_j\left(\mathbf{x}_{\mathcal{I}_j}\right),$$

Since that $\sum_{j \in \mathcal{S}_i} w^y_j f_j\left(\mathbf{x}_{\mathcal{I}_j}\right)$ is not a function of $x_i$, we have that $\varphi(\cdot)$ is a function without the interaction $\mathcal{I}$, which contradicts our assumption. $\qquad\square$

The reverse of this statement, that a common descendant will create an interaction among input features, holds true in most cases. The existence of counterexamples is manifested when early hidden layers capture an interaction that is negated in later layers. For example, the effects of two interactions may be directly removed in the next layer, as in the case of the following expression: $\max\{w_1 x_1 + w_2 x_2, 0\} - \max\{-w_1 x_1 - w_2 x_2, 0\} = w_1 x_1 + w_2 x_2$. Such an counterexample is legitimate; however, due to random fluctuations, it is highly unlikely in practice that the $w_1$s and the $w_2$s from the left hand side are exactly equal.

# B   PAIRWISE INTERACTION STRENGTH VIA QUADRATIC APPROXIMATION

We can provide a finer interaction strength analysis on a bivariate ReLU function: $\max\{\alpha_1 x_1 + \alpha_2 x_2, 0\}$, where $x_1, x_2$ are two variables and $\alpha_1, \alpha_2$ are the weights for this simple network. We quantify the strength of the interaction between $x_1$ and $x_2$ with the cross-term coefficient of the best quadratic approximation. That is,

$$\beta_0, \ldots, \beta_5 = \operatorname*{argmin}_{\beta_i, i=0,\ldots,5} \iint_{-1}^{1} \left[\beta_0 + \beta_1 x_1 + \beta_2 x_2 + \beta_3 x_1^2 + \beta_4 x_2^2 + \beta_5 x_1 x_2\right.$$
$$\left. - \max\{\alpha_1 x_1 + \alpha_2 x_2, 0\}\right]^2 dx_1\, dx_2.$$

Then for the coefficient of interaction $\{x_1, x_2\}$, $\beta_5$, we have that,

$$|\beta_5| = \frac{3}{4}\left(1 - \frac{\min\{\alpha_1^2, \alpha_2^2\}}{5\max\{\alpha_1^2, \alpha_2^2\}}\right)\min\{|\alpha_1|, |\alpha_2|\}. \tag{2}$$

Note that the choice of the region $(-1, 1) \times (-1, 1)$ is arbitrary: for larger region $(-c, c) \times (-c, c)$ with $c > 1$, we found that $|\beta_5|$ scales with $c^{-1}$. By the results of Proposition B, the strength of the interaction can be well-modeled by the minimum value between $|\alpha_1|$ and $|\alpha_2|$. Note that the factor before $\min\{|\alpha_1|, |\alpha_2|\}$ in Equation (2) is almost a constant with less than 20% fluctuation.

## C    PROOF FOR LEMMA 3

**Lemma 3** (Neural Network Lipschitz Estimation). *Let the activation function $\phi(\cdot)$ be a 1-Lipschitz function. Then the output $y$ is $z_i^{(\ell)}$-Lipschitz with respect to $h_i^{(\ell)}$.*

*Proof.* For non-differentiable $\phi(\cdot)$ such as the ReLU function, we can replace it with a series of differentiable 1-Lipschitz functions that converges to $\phi(\cdot)$ in the limit. Therefore, without loss of generality, we assume that $\phi(\cdot)$ is differentiable with $|\partial_x \phi(x)| \leq 1$. We can take the partial derivative of the final output with respect to $h_i^{(\ell)}$, the $i$-th unit at the $\ell$-th hidden layer:

$$\frac{\partial y}{\partial h_i^{(\ell)}} = \sum_{j_{\ell+1}, \ldots, j_L} \frac{\partial y}{\partial h_{j_L}^{(L)}} \frac{\partial h_{j_L}^{(L)}}{\partial h_{j_{L-1}}^{(L-1)}} \cdots \frac{\partial h_{j_{\ell+1}}^{(\ell+1)}}{\partial h_i^{(\ell)}}$$
$$= \mathbf{w}^{y\top} \mathrm{diag}(\dot{\boldsymbol{\phi}}^{(L)}) \mathbf{W}^{(L)} \cdots \mathrm{diag}(\dot{\boldsymbol{\phi}}^{(\ell+1)}) \mathbf{W}^{(\ell+1)},$$

where $\dot{\boldsymbol{\phi}}^{(\ell)} \in \mathbb{R}^{p_\ell}$ is a vector that

$$\dot{\phi}_k^{(\ell)} = \partial_x \phi \left( \mathbf{W}_{k,:}^{(\ell)} \mathbf{h}^{(\ell-1)} + b_k^{(\ell)} \right).$$

We can conclude the Lemma by proving the following inequality:

$$\left| \frac{\partial y}{\partial h_i^{(\ell)}} \right| \leq |\mathbf{w}^y|^\top \left| \mathbf{W}^{(L)} \right| \cdots \left| \mathbf{W}_{:,i}^{(\ell+1)} \right| = z_i^{(\ell)}.$$

The left-hand side can be re-written as

$$\sum_{j_{\ell+1}, \ldots, j_L} w_{j_L}^y \dot{\phi}_{j_L}^{(L)} W_{j_L, j_{L-1}}^{(L)} \dot{\phi}_{j_{L-1}}^{(L-1)} \cdots \dot{\phi}_{j_{\ell+1}}^{(\ell+1)} W_{j_{\ell+1}, i}^{(\ell+1)}.$$

The right-hand side can be re-written as

$$\sum_{j_{\ell+1}, \ldots, j_L} |w_{j_L}^y| \left| W_{j_L, j_{L-1}}^{(L)} \right| \cdots \left| W_{j_{\ell+1}, i}^{(\ell+1)} \right|.$$

We can conclude by noting that $|\partial_x \phi(x)| \leq 1$. $\qquad\qquad\square$

## D    PROOF FOR THEOREM 4

**Theorem 4** (Improving the ranking of higher-order interactions). *Let $\mathcal{R}$ be the set of interactions proposed by Algorithm 1 with $\mu(\cdot) = \min(\cdot)$, let $\mathcal{I} \in \mathcal{R}$ be a $d$-way interaction where $d \geq 3$, and let $\mathcal{S}$ be the set of subset $(d-1)$-way interactions of $\mathcal{I}$ where $|\mathcal{S}| = d$. Assume that for any hidden unit $j$ which proposed $s \in \mathcal{S} \cap \mathcal{R}$, $\mathcal{I}$ will also be proposed at the same hidden unit, and $\omega_j(\mathcal{I}) > \frac{1}{d}\omega_j(s)$. Then, one of the following must be true: a) $\exists s \in \mathcal{S} \cap \mathcal{R}$ ranked lower than $\mathcal{I}$, i.e., $\omega(\mathcal{I}) > \omega(s)$, or b) $\exists s \in \mathcal{S}$ where $s \notin \mathcal{R}$.*

*Proof.* Suppose for the purpose of contradiction that $\mathcal{S} \subseteq \mathcal{R}$ and $\forall s \in \mathcal{S}$, $\omega(s) \geq \omega(\mathcal{I})$. Because $\omega_j(\mathcal{I}) > \frac{1}{d}\omega_j(s)$,

$$\omega(\mathcal{I}) = \sum_{s \in \mathcal{S} \cap \mathcal{R}} \sum_{j \text{ propose } s} z_j \omega_j(\mathcal{I}) > \frac{1}{d} \sum_{s \in \mathcal{S} \cap \mathcal{R}} \sum_{j \text{ propose } s} z_j \omega_j(s) = \frac{1}{d} \sum_{s \in \mathcal{S} \cap \mathcal{R}} \omega(s).$$

Since $\forall s \in \mathcal{S}$, $\omega(s) \geq \omega(\mathcal{I})$,

$$\frac{1}{d} \sum_{s \in \mathcal{S} \cap \mathcal{R}} \omega(s) \geq \frac{1}{d} \sum_{s \in \mathcal{S} \cap \mathcal{R}} \omega(\mathcal{I})$$

Since $\mathcal{S} \subseteq \mathcal{R}$, $|\mathcal{S} \cap \mathcal{R}| = d$. Therefore,

$$\frac{1}{d} \sum_{s \in \mathcal{S} \cap \mathcal{R}} \omega(\mathcal{I}) \geq \frac{1}{d} \omega(\mathcal{I}) d \geq \omega(\mathcal{I}),$$

which is a contradiction.

$\square$

## E    ROC CURVES

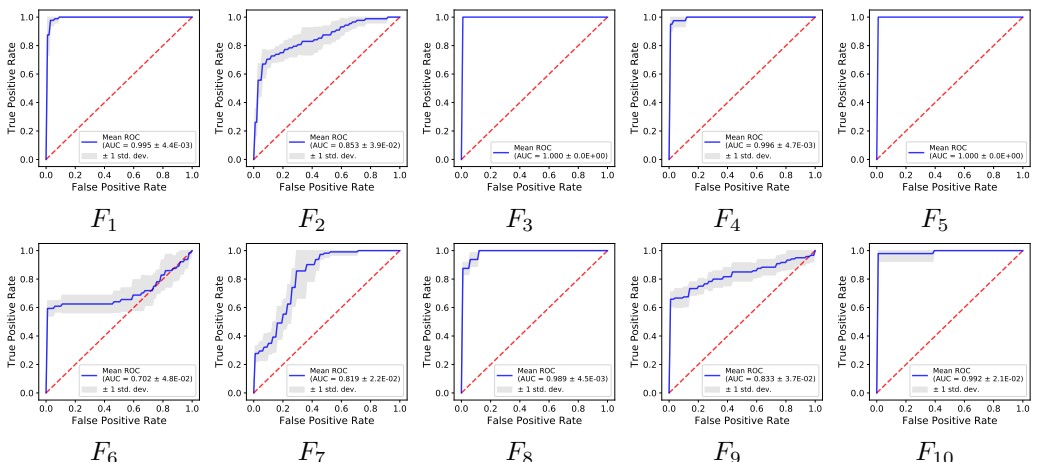

Figure 9: ROC curves of NID-*MLP-M* corresponding to Table 2

## F    LARGE $p$ EXPERIMENT

We evaluate our approach in a large $p$ setting with pairwise interactions using the same synthetic function as in Purushotham et al. (2014). Specifically, we generate a dataset of $n$ samples and $p$ features $\{(\mathbf{X}^{(i)}, y^{(i)})\}$ using the function

$$y^{(i)} = \boldsymbol{\beta}^\top \mathbf{X}^{(i)} + \mathbf{X}^{(i)\top} \mathbf{W} \mathbf{X}^{(i)} + \epsilon^{(i)},$$

where $\mathbf{X}^{(i)} \in \mathbb{R}^p$ is the $i^{th}$ instance of the design matrix $\mathbf{X} \in \mathbb{R}^{p \times n}$, $y^{(i)} \in \mathbb{R}$ is the $i^{th}$ instance of the response variable $\mathbf{y} \in \mathbb{R}^{n \times 1}$, $\mathbf{W} \in \mathbb{R}^{p \times p}$ contains the weights of pairwise interactions, $\boldsymbol{\beta} \in \mathbb{R}^p$ contains the weights of main effects, $\epsilon^{(i)}$ is noise, and $i = 1, \ldots, n$. $\mathbf{W}$ was generated as a sum of $K$ rank one matrices, $\mathbf{W} = \sum_{k=1}^{K} \boldsymbol{a}_k \boldsymbol{a}_k^\top$.

In this experiment, we set $p = 1000$, $n = 1\text{e}4$, and $K = 5$. $\mathbf{X}$ is normally distributed with mean 0 and variance 1, and $\epsilon^{(i)}$ is normally distributed with mean 0 and variance 0.1. Both $\boldsymbol{a}_k$ and $\boldsymbol{\beta}$ are sparse vectors of 2% nonzero density and are normally distributed with mean 0

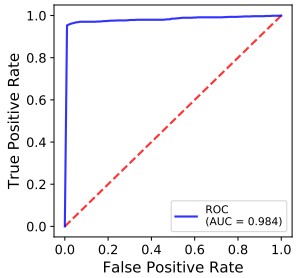

Figure 10: ROC curve for large $p$ experiment

and variance 1. We train *MLP-M* with the same hyperparameters as before (Section 5.1) but with a larger main network architecture of five hidden layers, with first-to-last layers sizes of 500, 400, 300, 200, and 100. Interactions are then extracted using the `NID` framework.

From this experiment, we obtain a pairwise interaction strength AUC of 0.984 on 950 ground truth pairwise interactions, where the AUC is measured in the same way as those in Table 2. The corresponding ROC curve is shown in Figure 10.

## G  COMPARING REGULARIZATION METHODS

We compare the average performance of `NID` for different regularizations on *MLP-M* networks. Specifically, we compare interaction detection performance when an *MLP-M* network has L1, L2, or group lasso regularization[4]. While L1 and L2 are common methods for regularizing neural network weights, group lasso is used to specifically regularize groups in the input weight matrix because weight connections into the first hidden layer are especially important in this work. In particular, we study group lasso by 1) forming groups associated with input features, and 2) forming groups associated with hidden units in the input weight matrix. In this experimental setting, group lasso effectively conducts variable selection for associated groups. Scardapane et al. (2017) define *group lasso* regularization for neural networks in Equation 5. Denote group lasso with input groups a $R_{\mathrm{GL}}^{(i)}$ and group lasso with hidden unit groups as $R_{\mathrm{GL}}^{(h)}$. In order to apply both group and individual level sparsity, Scardapane et al. (2017) further define *sparse group lasso* in Equation 7. Denote sparse group lasso with input groups as $R_{\mathrm{SGL}}^{(i)}$ and sparse group lasso with hidden unit groups as $R_{\mathrm{SGL}}^{(h)}$.

Networks that had group lasso or sparse group lasso applied to the input weight matrix had L1 regularization applied to all other weights. In our experiments, we use large dataset sizes of 1e5 and tune the regularizers by gradually increasing their respective strengths from zero until validation performance worsens from underfitting. The rest of the neural network hyperparameters are the same as those discussed in Section 5.1. In the case of the group lasso and sparse group lasso experiments, L1 norms were tuned the same as in the standard L1 experiments.

In Table 4, we report average pairwise interaction strength AUC over 10 trials of each function in our synthetic test suite (Table 1) for the different regularizers.

Table 4: Average AUC of pairwise interaction strengths proposed by `NID` for different regularizers. Evaluation was conducted on the test suite of synthetic functions (Table 1).

|  | L1 | L2 | $R_{\mathrm{GL}}^{(i)}$ | $R_{\mathrm{GL}}^{(h)}$ | $R_{\mathrm{SGL}}^{(i)}$ | $R_{\mathrm{SGL}}^{(h)}$ |
|---|---|---|---|---|---|---|
| average | $0.94 \pm 2.9\mathrm{e}{-2}$ | $0.94 \pm 2.5\mathrm{e}{-2}$ | $0.95 \pm 2.4\mathrm{e}{-2}$ | $0.94 \pm 2.5\mathrm{e}{-2}$ | $0.93 \pm 3.2\mathrm{e}{-2}$ | $0.94 \pm 3.0\mathrm{e}{-2}$ |

## H  COMPARISONS WITH LOGISTIC REGRESSION BASELINES

We perform experiments with our `NID` approach on synthetic datasets that have binary class labels as opposed to continuous outcome variables (e.g. Table 1). In our evaluation, we compare our method against two logistic regression methods for multiplicative interaction detection, *Factorization Based High-Order Interaction Model (FHIM)* (Purushotham et al., 2014) and *Sparse High-Order Logistic Regression (Shooter)* (Min et al., 2014). In both comparisons, we use dataset sizes of $p = 200$ features and $n = 1\mathrm{e}4$ samples based on *MLP-M*'s fit on the data and the performance of the baselines. We also make the following modifications to *MLP-M* hyperparameters based on validation performance: the main *MLP-M* network has first-to-last layer sizes of 100, 60, 20 hidden units, the univariate networks do not have any hidden layers, and the L1 regularization constant is set to $5\mathrm{e}{-4}$. All other hyperparameters are kept the same as in Section 5.1.

**FHIM** Purushotham et al. (2014) developed a feature interaction matrix factorization method with L1 regularization, *FHIM*, that identifies pairwise multiplicative interactions in flexible $p$ settings.

---

[4]We omit dropout in these experiments because it significantly lowered the predictive performance of *MLP-M* in our regression evaluation.

When used in a logistic regression model, *FHIM* detects pairwise interactions that are predictive of binary class labels. For this comparison, we used data generated by Equation 5 in Purushotham et al. (2014), with $K = 2$ and sparsity factors being $5\%$ to generate 73 ground truth pairwise interactions.

In Table 5, we report average pairwise interaction detection AUC over 10 trials, with a maximum and a minimum AUC removed.

Table 5: Pairwise AUC comparison between *FHIM* and `NID` with $p = 200$, $n = 1\mathrm{e}4$

|  | *FHIM* | NID |
|---|---|---|
| average | $0.925 \pm 2.3\mathrm{e}{-3}$ | $0.973 \pm 6.1\mathrm{e}{-3}$ |

**Shooter** Min et al. (2014) developed *Shooter*, an approach of using a tree-structured feature expansion to identify pairwise and higher-order multiplicative interactions in a L1 regularized logistic regression model. This approach is special because it relaxes our hierarchical hereditary assumption, which requires subset interactions to exist when their corresponding higher-order interaction also exists (Section 3). Specifically, *Shooter* relaxes this assumption by only requiring *at least one* $(d - 1)$-way interaction to exist when its corresponding $d$-way interaction exists.

With this relaxed assumption, *Shooter* can be evaluated in depth per level of interaction order. We compare `NID` and *Shooter* under the relaxed assumption by also evaluating `NID` per level of interaction order, where Algorithm 1 is specifically being evaluated. We note that our method of finding a cutoff on interaction rankings (Section 4.3) strictly depends on the hierarchical hereditary assumption both within the same interaction order and across orders, so instead we set cutoffs by thresholding the interaction strengths by a low value, $1\mathrm{e}{-3}$.

For this comparison, we generate and consider interaction orders up to degree 5 (5-way interaction) using the procedure discussed in Min et al. (2014), where the interactions do not have strict hierarchical correspondence. We do not extend beyond degree 5 because *MLP-M*'s validation performance begins to degrade quickly on the generated dataset. The sparsity factor was set to be $5\%$, and to simplify the comparison, we did not add noise to the data.

In Table 6, we report precision and recall scores of *Shooter* and `NID`, where the scores for `NID` are averaged over 10 trials. While *Shooter* performs near perfectly, `NID` obtains fair precision scores but generally low recall. When we observe the interactions identified by `NID` per level of interaction order, we find that the interactions across levels are always subsets or supersets of another predicted interaction. This strict hierarchical correspondence would inevitably cause `NID` to miss many true interactions under this experimental setting. The limitation of *Shooter* is that it must assume the form of interactions, which in this case is multiplicative.

Table 6: Precision and recall of interaction detection per interaction order, where $|\mathcal{I}|$ denotes interaction cardinality. The *Shooter* implementation is deterministic.

| $|\mathcal{I}|$ | *Shooter* precision | *Shooter* recall | NID precision | NID recall |
|---|---|---|---|---|
| 2 | $87.5\%$ | $100\%$ | $90\% \pm 11\%$ | $21\% \pm 9.6\%$ |
| 3 | $96.0\%$ | $96.0\%$ | $91\% \pm 8.4\%$ | $19\% \pm 8.5\%$ |
| 4 | $100\%$ | $100\%$ | $60\% \pm 13\%$ | $21\% \pm 9.3\%$ |
| 5 | $100\%$ | $100\%$ | $73\% \pm 8.4\%$ | $30\% \pm 13\%$ |

## I   SPURIOUS MAIN EFFECT APPROXIMATION

In the synthetic function $F_6$ (Table 2), the $\{8, 9, 10\}$ interaction, $\sqrt{x_8^2 + x_9^2 + x_{10}^2}$, can be approximated as main effects for each variable $x_8$, $x_9$, and $x_{10}$ when at least one of the three variables is close to $-1$ or $1$. Note that in our experiments, these variables were uniformly distributed between $-1$ and $1$.

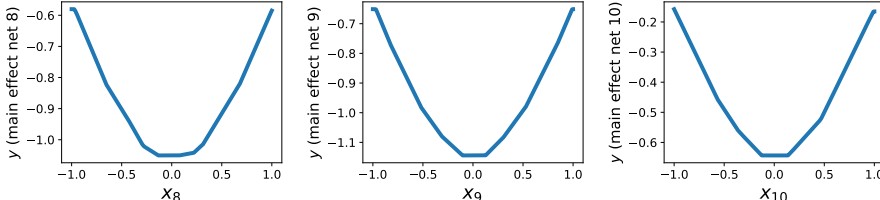

Figure 11: Response plots of an *MLP-M*'s univariate networks corresponding to variables $x_8$, $x_9$, and $x_{10}$. The *MLP-M* was trained on data generated from synthetic function $F_6$ (Table 2). Note that the plots are subject to different levels of bias from the *MLP-M*'s main multivariate network.

For example, let $x_{10} = 1$ and $z^2 = x_8^2 + x_9^2$, then by taylor series expansion at $z = 0$,

$$\sqrt{z^2 + 1} \approx 1 + \frac{1}{2}z^2 = 1 + \frac{1}{2}x_8^2 + \frac{1}{2}x_9^2.$$

By symmetry under the assumed conditions,

$$\sqrt{x_8^2 + x_9^2 + x_{10}^2} \approx c + \frac{1}{2}x_8^2 + \frac{1}{2}x_9^2 + \frac{1}{2}x_{10}^2,$$

where $c$ is a constant.

In Figure 11, we visualize the $x_8$, $x_9$, $x_{10}$ univariate networks of a *MLP-M* (Figure 2) that is trained on $F_6$. The plots confirm our hypothesis that the *MLP-M* models the $\{8,9,10\}$ interaction as spurious main effects with parabolas scaled by $\frac{1}{2}$.

## J    INTERACTION VISUALIZATION

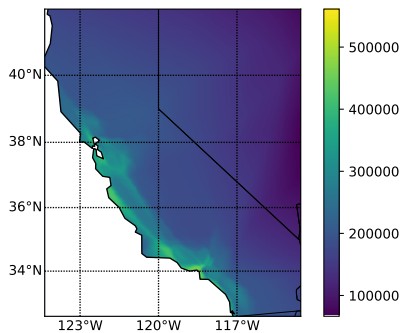

Figure 12: A heatmap showing relative housing prices in California based on longitude and latitude.

In Figure 12, we visualize the interaction between longitude and latitude for predicting relative housing price in California. This visualization is extracted from the longitude-latitude interaction network within *MLP-Cutoff*[5], which was trained on the cal housing dataset (Pace & Barry, 1997). We can see that this visualization cannot be generated by longitude and latitude information in additive form, but rather the visualization needs special joint information from both features. The highly interacting nature between longitude and latitude confirms the high rank of this interaction in our `NID` experiments (see the $\{1, 2\}$ interaction for cal housing in Figures 5 and 7).

---

[5]*MLP-Cutoff* is a generalization of GA$^2$Ms, which are known as intelligible models (Lou et al., 2013).

