# OpenReview forum: "Detecting Statistical Interactions from Neural Network Weights"
_ICLR.cc/2018/Conference — Accept (Poster)_

### Official Review · AnonReviewer3 · 2017-11-27
**An interesting use case of neural networks for a traditional statistical problem!**

**Rating:** 7
**Confidence:** 4

**Review:**

This paper develops a novel method to use a neural network to infer statistical interactions between input variables without assuming any explicit interaction form or order. First the paper describes that an 'interaction strength' would be captured through a simple multiplication of the aggregated weight and the weights of the first hidden layers. Then, two simple networks for the main and interaction effects are modeled separately, and learned jointly with posing L1-regularization only on the interaction part to cancel out the main effect as much as possible. The automatic cutoff determination is also proposed by using a GAM fitting based on these two networks. A nice series of experimental validations demonstrate the various types of interactions can be detected, while it also fairly clarifies the limitations.

In addition to the related work mentioned in the manuscript, interaction detection is also originated from so-called AID, literally intended for 'automatic interaction detector' (Morgan & Sonquist, 1963), which is also the origin of CHAID and CART, thus the tree-based methods like Additive Groves would be the one of main methods for this. But given the flexibility of function representations, the use of neural networks would be worth rethinking, and this work would give one clear example.

I liked the overall ideas which is clean and simple, but also found several points still confusing and unclear.

1) One of the keys behind this method is the architecture described in 4.1. But this part sounds quite heuristic, and it is unclear to me how this can affect to the facts such as Theorem 4 and Algorithm 1. Absorbing the main effect is not critical to these facts? In a standard sense of statistics, interaction would be something like residuals after removing the main (additive) effect. (like a standard test by a likelihood ratio test for models with vs without interactions)

2) the description about the neural network for the main effect is a bit unclear. For example, what does exactly mean the 'networks with univariate inputs for each input variable'? Is my guessing that it is a 1-10-10-10-1 network (in the experiments) correct...? Also, do g_i and g_i' in the GAM model (sec 4.3) correspond to the two networks for the main and interaction effects respectively?

3) mu is finally fixed at min function, and I'm not sure why this is abstracted throughout the manuscript. Is it for considering the requirements for any possible criteria?

Pros:
- detecting (any order / any form of) statistical interactions by neural networks is provided.
- nice experimental setup and evaluations with comparisons to relevant baselines by ANOVA, HierLasso, and Additive Groves.

Cons:
- some parts of explanations to support the idea has unclear relationship to what was actually done, in particular, for how to cancel out the main effect.
- the neural network architecture with L1 regularization is a bit heuristic, and I'm not surely confident that this architecture can capture only the interaction effect by cancelling out the main effect.

---

> ### Author Response · Authors · 2017-12-14
> **authors' response**
>
> Thank you for your comments and suggestions.
>
> >> “it is unclear to me how this can affect to the facts such as Theorem 4 and Algorithm 1.”
> The existence of the univariate networks does not affect Algorithm 1 or Theorem 4. The univariate networks are meant to reduce the modeling of spurious interactions in the main fully-connected network to improve interaction detection performance.
>
> >>  On the architecture of the univariate and GAM networks
> Your understanding is correct.
>
> >> Regarding the abstraction of mu,
> We did not define mu until later in the paper because mu was determined by experiments

---

### Official Review · AnonReviewer1 · 2017-11-27
**Interesting method, more work to do on the experimental side**

**Rating:** 7
**Confidence:** 4

**Review:**

This paper presents a method to identify high-order interactions from the weights of feedforward neural networks.

The main benefits of the method are:
1)	Can detect high order interactions and there’s no need to specify the order (unlike, for example, in lasso-based methods).
2)	Can detect interactions appearing inside of non-linear function (e.g. sin(x1 * x2))

The method is interesting, in particular if benefit #2 holds experimentally. Unfortunately, there are too many gaps in the experimental evaluation of this paper to warrant this claim right now.

Major:

1)	Arguably, point 1 is not a particularly interesting setting. The order of the interactions tested is mainly driven by the sample size of the dataset considered, so in some sense the inability to restrict the order of the interaction found can actually be a problem in real settings.
Because of this, it would be very helpful to separate the evaluation of benefit 1 and 2 at least in the simulation setting. For example, simulate a synthetic function with no interactions appearing in non-linearities (e.g. x1+x2x3x4+x4x6) and evaluate the different methods at different sample sizes (e.g. 100 samples to 1e5 samples). The proposed method might show high type-1 error under this setting. Do the same for the synthetic functions already in the paper. By the way, what is the sample size of the current set of synthetic experiments?
2)	The authors claim that the proposed method identifies interactions “without searching an exponential solution space of possible interactions”. This is misleading, because the search of the exponential space of interactions happens during training by moving around in the latent space identified by the intermediate layers. It could perhaps be rephrased as “efficiently”.
3)	It’s not clear from the text whether ANOVA and HierLasso are only looking for second order interactions. If so, why not include a lasso with n-order interactions as a baseline?
4)	Why aren’t the baselines evaluated on the real datasets and heatmaps similar to figure 5 are produced?
5)	Is it possible to include the ROC curves corresponding to table 2?


Minor:

1)	Have the authors thought about statistical testing in this framework? The proposed method only gives a ranking of possible interactions, but does not give p-values or similar (e.g. FDRs).
2)	12 pages of text. Text is often repetitive and can be shortened without loss of understanding or reproducibility.

---

> ### Author Response · Authors · 2017-12-14
> **rebuttal**
>
> Thank you for your comments and suggestions. We conducted some experiments based on your suggestions and provide our responses to your major points below. We have also included additional results in Appendices E and F.
>
> 1) Our proposed method strongly relies on the assumption that the neural network fit well to data, since we are extracting interactions from the learned network weights. The number of data points available plays a critical role here because a small amount of data can cause the neural network to overfit, causing our method to miss true interactions and find spurious ones instead. To avoid this scenario, we employed modern tricks to help our neural network fit well to the data (e.g. early stopping, regularization).  That being said, we advise against using our framework when the number of data samples, n, is too small for normal neural networks, e.g., when n < p, where p is the number of features. Under such scenarios, one might need to impose much stronger assumptions on the data, which goes against our proposal of a general interaction detection algorithm.
>
> For assurance, we conducted the experiments that you requested and confirmed that our approach does well on the multiplicative synthetic function x1+x2x3x4+x4x6 for datasets of sizes [1e2, 1e3, 1e4, 1e5], obtaining average interaction ranking AUCs of [0.99, 1.0, 1.0, 1.0], respectively. The average AUCs for our 10 synthetic functions with nonlinearities (combined) are [0.57,0.83,0.92,0.94] respectively. The baseline methods are specified to find multiplicative interactions, so their AUC is 1 for the multiplicative synthetic function. Note that we can only obtain interactions accurately when there is enough data to train the model, as seen in the improving scores with more data samples.  In our synthetic experiments, we used 10k training samples (and 10k valid/10k test), and this has been updated in our paper. We also updated our paper with a large-p experiment on multiplicative interactions in Appendix F, where we obtained an AUC of 0.98.
>
> 2) While the neural network is technically searching interactions during training, the cost of this implicit exponential search is faster than an explicit exponential search of the space of interaction candidates. Our work avoids this explicit search.
>
> 3) We originally did not include higher-order detection experiments with ANOVA and Lasso because they are mis-specified to handle detecting the general non-additive form of interactions. For assurance, we ran experiments on ANOVA and Lasso and got average top-rank recall scores of 0.47 and 0.44 respectively, which are much lower than the 0.65 average obtained by our approach.

---

### Official Review · AnonReviewer2 · 2017-11-28
**Feature interaction identification by multiplying |neural network weight matrices|**

**Rating:** 7
**Confidence:** 5

**Review:**

Based on a hierarchical hereditary assumption, this paper identifies pairwise and high-order feature interactions by re-interpreting neural network weights, assuming higher-order interactions exist only if all its induced lower-order interactions exist. Using a multiplication of the absolute values of all neural network weight matrices on top of the first hidden layer, this paper defines the aggregated strength z_r of each hidden unit r contributing to the final target output y. Multiplying z_r by some statistics of weights connecting a subset of input features to r and summing over r results in final interaction strength of each feature interaction subsets, with feature interaction order equal to the size of each feature subset.

Main issues:

1. Aggregating neural network weights to identify feature interactions is very interesting. However, completely ignoring
activation functions makes the method quite crude.

2. High-order interacting features must share some common hidden unit somewhere in a hidden layer within a deep neural network. Restricting to the first hidden layer in Algorithm 1 inevitably misses some important feature interactions.

3. The neural network weights heavily depends on the l1-regularized neural network training, but a group lasso penalty makes much more sense. See Group Sparse Regularization for Deep Neural Networks (https://arxiv.org/pdf/1607.00485.pdf).

4. The experiments are only conducted on some synthetic datasets with very small feature dimensionality p. Large-scale experiments are needed.

5. There are some important references missing. For example, RuleFit is a good baseline method for identifying feature interactions based on random forest and l1-logistic regression (Friedman and Popescu, 2005, Predictive learning via rule ensembles); Relaxing strict hierarchical hereditary constraints, high-order l1-logistic regression based on tree-structured feature expansion identifies pairwise and high-order multiplicative feature interactions (Min et al. 2014, Interpretable Sparse High-Order Boltzmann Machines); Without any hereditary constraint, feature interaction matrix factorization with l1 regularization identifies pairwise feature interactions on datasets with high-dimensional features (Purushotham et al. 2014, Factorized Sparse Learning Models with Interpretable High Order Feature Interactions).

6. At least, RuleFit (Random Forest regression for getting rules + l1-regularized regression) should be used as a baseline in the experiments.

Minor issues:

Ranking of feature interactions in Algorithm 1 should be explained in more details.

On page 3: b^{(l)} \in R^{p_l}, l should be from 1, .., L. You have b^y.


In summary, the idea of using neural networks for screening pairwise and high-order feature interactions is novel, significant, and interesting.  However, I strongly encourage the authors to perform additional experiments with careful experiment design to address some common concerns in the reviews/comments for the acceptance of this paper.

========
The additional experimental results are convincing, so I updated my rating score.

---

> ### Author Response · Authors · 2017-12-14
> **rebuttal**
>
> Thank you for your comments and suggestions to improve the paper. Below are our responses to the main points of your comments:
>
> 1. >> “However, completely ignoring activation functions makes the method quite crude.”
> Our approach to aggregating weight matrices depends on the activation functions in two ways: 1) the use of matrix multiplications is based on Lemma 3, which depends on the activation functions being 1-Lipschitz, and 2) the averaging of weights is empirically determined from neural networks with ReLU activation.
>
> 2. >> “Restricting to the first hidden layer in Algorithm 1 inevitably misses some important feature interactions.”
> This is an interesting point. We did consider it before. However, it is not straightforward how to incorporate the idea of common hidden units at intermediate layers to get better interaction detection performance. Our previous studies show that naively using the intermediate hidden layers to suggest new interactions have resulted in worse performance in interaction detection because the connections between input features and intermediate layers are not direct.
>
> 3. >> “a group lasso penalty makes much more sense”
> In general, group lasso requires specifying groupings a priori. It is unclear how to tailor the group lasso penalty to discover interactions, but group lasso might offer an alternative way of finding a cutoff on interaction rankings.
>
> 4. >> “Large-scale experiments are needed.”
> We have conducted experiments with large scale p (p=1000, 950 pairwise interactions) as you suggested and obtained a pairwise interaction strength AUC of 0.984. The full experimental setting can be found in our updated paper in Appendix F, which follows Purushotham et al. 2014 on how to generate large p noisy data.
>
> 5 & 6. >> “, RuleFit should be used as a baseline in the experiments.”
> We have added experiments with RuleFit into Table 2 as you suggested. Our approach outperforms RuleFit. This is consistent with previous work by Lou et al. 2013, “Accurate and Intelligible Models with Pairwise Interactions”, which found that RuleFit did not perform better than Additive Groves, our main baseline.

---

> > ### Comment · AnonReviewer2 · 2017-12-16
> > **Group Lasso regularization is natural, important, and necessary**
> >
> > The neural network weights heavily depends on the l1-regularized neural network training, for which a group lasso penalty makes much more sense.
> >
> > Since this paper focuses on the first hidden layer, the connection weights between input features and the hidden units in the first hidden layer are highly important.
> >
> > On one hand, to enforce sparsity,  connection weights from each individual input feature to all the hidden units naturally form a group. If there are $n$ features, there will be $n$ groups of weights connecting input features and the first hidden layer, keeping only important features for interaction detection. Other weights in higher layers can be regularized with standard Lasso. At least intuitively, Group Lasso based on this natural grouping should work much better than a standard Lasso for eliminating false positive feature interactions.
> >
> > On the other hand, considering that the number of hidden units in the first layer is loosely set based on validations, weights connecting all input features to each first-layer hidden unit also naturally form a group, which renders the neural network only keeping highly important competitive hidden units for interaction identification.
> >
> > Therefore, experiments on interaction identification with Group Lasso regularization is natural, intuitive, important, and necessary for the proposed method.

---

> > > ### Author Response · Authors · 2017-12-20
> > > **authors' response**
> > >
> > > Thank you for clarifying your suggestion on group lasso. We have conducted experiments on group lasso as you suggested. After tuning regularization strengths, we found that indeed the average AUC of group lasso with input groups is slightly better than vanilla lasso, but the difference is not statistically significant. The details of additional experiments with group lasso are included in Appendix G.
> > >
> > > We also plan to share our code in the future so that readers can replicate the experiments.

---

> > > > ### Comment · AnonReviewer2 · 2018-01-03
> > > > **The additional experiments are convincing**
> > > >
> > > > The revision is convincing, so the rating score is updated.

---

> ### Author Response · Authors · 2017-12-28
> **rebuttal (cont'd)**
>
> 5 & 6. >> On the remaining baseline references
>
> We have added references for the remaining baselines as you suggested (on Page 2 and in Appendix H). The references are Interpretable Sparse High-Order Boltzmann Machines, Min et al. 2014 and Factorized Sparse Learning Models with Interpretable High Order Feature Interactions, Purushotham et al. 2014.
>
> We performed experiments comparing our method to the "Shooter" and "FHIM" baselines from the references. The details and results of our experiments are shown in Appendix H.  We found that indeed, the Shooter baseline benefits from relaxing strict hierarchical hereditary constraints

---

### Decision · Program_Chairs · 2018-01-29
**ICLR 2018 Conference Acceptance Decision**

**Decision:**

Accept (Poster)

**Comment:**

The paper proposes a way of detecting statistical interactions in a dataset based on the weights learned by a DNN. The idea is interesting and quite useful as is showcased in the experiments. The reviewers feel that the paper is also quite well written and easy to follow.